# LncRNA HOTAIR promotes proliferation, invasion and migration in NSCLC cells via the CCL22 signaling pathway

Hanlin Liang◉*, Jiewen Peng◉

Chemotherapy Department, Zhongshan City People's Hospital, Zhongshan, Guangdong Province, People's Republic of China

* lianghanlin@gmai.com

## Abstract

Long noncoding RNA (LncRNA) is a new type of regulatory RNA. LncRNA HOX antisense intergenic RNA (HOTAIR), as an oncogene in non-small cell lung cancer (NSCLC), is one of the key determinants of tumor progression. However, its possible molecular mechanism and the immunomodulatory pathway involved in NSCLC are still unclear. This study aims to explore whether HOTAIR promotes proliferation, migration and invasion of the NSCLC cells by inhibiting the expression of C-C Motif Chemokine Ligand 22 (CCL22). We collected 30 clinical samples of cancer and adjacent normal tissues from the patients with NSCLC, using real-time quantitative polymerase chain reaction (RT-qPCR) to detect the LncRNA HOTAIR and CCL22 mRNA expression in tissues. Immunohistochemistry was used to detect the protein expression of CCL22 in cancer and adjacent normal tissues. Cell experiments were conducted to verify that LncRNA HOTAIR regulates the expression of CCL22 and participates in the progress of NSCLC. The antisense oligonucleotide (ASO) probe interfering with LncRNA HOTAIR and the interference fragment of CCL22 (si-CCL22) were constructed. A549 cells were co-transfected with ASO-HOTAIR and si-CCL22. We used RT-qPCR to detect the expression of LncRNA HOTAIR and CCL22 mRNA in the cells, enzyme-linked immunosorbent assay (ELISA) used to detect the CCL22 protein level in the cell supernatant. 3-(4,5-dimethylthiazol-2-yl)-5-(3-carboxymethoxyphenyl)-2-(4-sulfophenyl)-2H-tetrazolium (MTS) assay was applied to detect cell proliferation, the Flow cytometry to detect cell apoptosis. Finally, the Transwell test was utilized to detect cell migration and invasion. In conclusion, this study suggests that HOTAIR may promote proliferation, migration and invasion of the NSCLC cells by inhibiting CCL22 expression, which may play a key role in NSCLC cell immunity.

## 1 Introduction

Lung cancer is a malignant tumor that seriously threatens human health and life nowadays, whose incidence is increasing in many countries [1]. Non-small cell lung cancer (NSCLC) is the most common histology in lung cancer, accounting for 80%-85% of lung cancer [2]. It is

**Data Availability Statement:** All relevant data are within the paper and its Supporting Information files.

**Funding:** This project is funded by Zhongshan Science and Technology Bureau. Hanlin Liang and

Jiewen Peng received the award. No conflicts of interest. The website of Zhongshan Science and Technology Bureau was listed as follow: http://kj. zs.gov.cn/ The funder had no role in study design, data collection and analysis, decision to publish, or preparation of the manuscript.

**Competing interests:** NO authors have competing interests.

generally believed that the pathogenesis of NSCLC is the result of a combination of factors such as heredity, immunity, environment, and living habits. Although a comprehensive treatment plan such as surgery, radiotherapy, and chemotherapy is used to treat early NSCLC, a large proportion of patients eventually die of local recurrence or distant metastasis, which makes it challenging to improve their overall survival rate.

A large number of studies have shown that the high expression of LncRNA HOTAIR is related to the poor prognosis of NSCLC, interference with HOTAIR expression inhibiting the proliferation and migration of the A549 cells [3]. Silencing HOTAIR can reduce EP4 protein levels and inhibit the growth of NSCLC cells, while overexpression of HOTAIR and SP1 can inhibit Xiaoji decoction (XJD)-decreased EP4 protein expression [4]. MicroRNA 221 (MiR-221) negatively regulates lncRNA HOTAIR to promote NSCLC cell apoptosis and can be used for the treatment of NSCLC [5]. HOTAIR is up-regulated in the NSCLC cells, regulating cell proliferation, migration, and invasion via miR-217/Dachshund homolog 1(DACH1) signaling pathway [6]. Simultaneously, HOTAIR inhibits the phosphorylation of ULK1 to reduce autophagy. Otherwise, silencing HOTAIR can lower the resistance of NSCLC cells to Crizotinib [7]. HOTAIR2, HOTAIR3 and HOTAIR5 promote the cell cycle through the restriction point G1-S phase by regulating the Rb-E2F pathway, affecting the proliferation, migration, and invasion in the non-small cell lung cancer cells through epithelial-mesenchymal transition (EMT) and β -Catenin pathway in vitro and in vivo [8]. In summary, these results indicate that HOTAIR promotes proliferation, migration, and invasion of the NSCLC cells, leading to poor prognosis. In addition, CCL22 and IL-37 inhibit the proliferation and EMT process of A549 cells [9]. The high expression of chemotactic factor CCL22 and CCL22 can recruit regulatory T cells (Treg) to the tumor site of NSCLC [10, 11]. These results suggest that CCL22 also inhibits the proliferation, migration, and invasion of the NSCLC cells. Therefore, this study aims to explore whether HOTAIR promotes the proliferation, migration, and invasion of the NSCLC cells by inhibiting the expression of CCL22 or not in the present study.

## 2 Materials and methods

### 2.1 Clinical verification of the correlation between LncRNA HOTAIR, CCL22 and NSCLC

We collected 30 pairs of clinical samples of cancer tissues and adjacent normal tissues acquired from NSCLC patients who underwent surgery in the Zhongshan City People's Hospital from July 1, 2019, to October 30, 2020. Then, RT-qPCR was used to detect the expression of LncRNA HOTAIR and CCL22 mRNA expression, immunohistochemistry utilized to test the protein expression of CCL22 in the tissue samples. Finally, the differential expressions of LncRNA HOTAIR and CCL22 mRNA between the NSCLC tissues and adjacent normal tissues were analyzed.

### 2.2 Cell culture

The A549 cells (purchased from ATCC) were inoculated into the DMEM medium (10% FBS and 1% dual antibody), cultured in a 37˚C, 5% CO2 incubator. When 70%~80% of the petri dish bottom was covered with cells, the cells were subcultured. Logarithmic stage cells were used for the follow-up tests in this study.

### 2.3 Cell transfection

The antisense oligonucleotide probe (ASO probe) interfering with LncRNA HOTAIR and the interference fragment of CCL22 (si-CCL22) were constructed, respectively. Logarithmic A549

cells were inoculated into 6-well plates (1.0×10 5 cells/well). After 24h culture, ASO-HOTAIR and si-CCL22 were co-transfected with Lipofectamine TM 2000 test box. The groups were listed as follows: (a) A549 cells+ASO-NC (Negative Control)+si-NC; (b) A549 cells+-ASO-HOTAIR+si-NC; (c) A549 cells+ASO-HOTAIR+si-CCL22. After transfection, the expressions of HOTAIR and CCL22 in cells were detected to verify the transfection effect, the cells used for the subsequent experiments.

## 2.4 Immunohistochemistry

The paraffin sections with 4μm were baked at 65˚C for 4h, dewaxed with xylene, then treated with alcohol. The slices were rinsed with water for 30min, the sodium citrate repair solution (PH = 6.0, o.01M) repaired by microwave with medium fire for 10min. Treated with 3% hydrogen peroxide solution for 15min, the tissues were blocked with 1% FBS for 30min. The primary anti-CCL22 antibody (1:200, ab9847, Abcam) was added to the sections, incubated at 4˚C overnight. The tissues added with the corresponding secondary antibody were incubated at room temperature for 1h. The sections were colored with DAB for 5min. Dehydrated with alcohol, the slices were sealed with neutral gum. The photos were captured with the microscope. Image-pro Plus 6.0 software was used to calculate the positive expression rate of the protein.

## 2.5 RT-qPCR

The RT-qPCR experiment was carried out in reference to the previous study [12]. The thoroughly ground tissues or cells were left for 10 min at 4˚C, centrifuged at 12 000 r/ min at 4˚C for 10 min. The supernatant being taken, total RNAs were extracted to determine the total concentration and purity of total RNA. The total reaction system was 20 μL, the sequence of primers shown in Table 1. The RT-qPCR procedures were as follows: ① Pre-denaturation at -95˚C for 5 min; ② 40 cycles of cyclic reaction—denaturation at 95˚C for 10 s, annealing at 60˚C for 20 s, extension at 72˚C for 40 s; (3) One cycle of the melting curve at -95˚C 15 s, 60˚C 60 s,95˚C 15 s. Three replicates were set for each sample. The mRNA expressions of HOTAIR and CCL22 in tissues and cells were calculated and analyzed by the method of 2 -ΔΔCT.

## 2.6 Flow cytometry to detect cell apoptosis

The transfected cells were cultured in 24-well plates for 24 h, with the initial number of cells in each group 5.0×10 4 cells/well. After culture, cells in each group were collected, apoptosis detected by Annexin V-FITC/PI kit.

## 2.7 ELISA to detect CCL22 concentration

The concentration of CCL22 in the supernatant was tested according to the instructions of ELISA kit operation (CUSABIO, CSB-E04660h). The prepared well plate was added with

**Table 1. Primer sequences.**

| Primer | | sequences |
|---|---|---|
| ASO-HOTAIR | sense | 5'– GAACGGGAGUACAGAGAGA –3 |
| si-CCL22 | sense | 5'– GCGUGGUGUUGCUAACCUUdTdT–3' |
| | antisense | 5'– AAGGUUAGCAACACCACGCdGdT–3' |
| HOTAIR | sense | 5'–A GGTAGAAAAAGCAACCACGAAGC –3' |
| | antisense | 5'– ACATAAACCTCTGTCTGTGAGTGCC –3' |
| H-CCL22 | sense | 5'– GAGATCTGTGCCGATCCCAG –3' |
| | antisense | 5'– AGGGAATGCAGAGAGTTGGC –3' |

100μL per well, incubated at 37˚C for 90 min. The biotin anti-human CCL22 antibody working solution of 100 μL was added to each well, reacted at 37˚C for 60 min. Then, each well was added with ABC working solution 100 μL, reacted at 37˚C for 30 min. The TMB color solution was added to the well for color rendering, the OD value measured at 450nm with a microplate reader. The standard curve was performed according to the MEASURED OD, and then the level of CCL22 protein was calculated.

### 2.8 MTS assay

The cell proliferation was detected by MTS as previous study [13]. In short, 4000 cells/well was seeded into a 96-well plate, MTS added for 30 minutes. The OD value at 490nm was detected with three replicates for each test.

### 2.9 Tanswell experiment testing cell migration and invasion

Migration experiment: cells were inoculated in the upper compartment of Transwell, with the initial number 5.0×10 4 cells/well. 500 μL complete medium was added to the lower chamber. After 24 h culture, the medium was abandoned, the upper chamber removed. The lower chamber cells were fixed with 4% paraformaldehyde, stained with 0.1% crystal violet. The cells were observed, counted under the microscope. Invasion test: The invasion procedure was consistent with the migration experiment except that Matrigel glue was pre-applied to the upper chamber of Transwell.

### 2.10 Statistical analysis

The data were expressed as mean ± standard deviation (SD), analyzed by SPSS 21.0 software. GraphPad Prism 6.0 was utilized to make the statistic charts. Comparison between the two groups was performed using the T-test of two independent samples. Univariate one-way analysis of variance (ANOVA) was used for comparison among the three groups, LST-t test used for pairwise comparison between groups. The correlation analysis between quantitative data is carried out by Person correlation analysis. $P < 0.05$ is considered statistically significant.

## 3 Results

### 3.1 The correlation between LncRNA HOTAIR, CCL22 in NSCLC

The RT-qPCR results showed that, the expression of LncRNA HOTAIR was up-regulated, while the expression of CCL22 mRNA was down-regulated in the non-small cell lung cancer clinical samples compared with adjacent normal tissues (Fig 1A and 1B). Pearson correlation analysis presented a strong negative correlation between LncRNA HOTAIR expression and CCL22 mRNA expression in the clinical samples of NSCLC (r = -0.7047, $P < 0.0001$) (Fig 1C). Immunohistochemical results (Fig 1D) showed that the CCL22 protein expression was significantly decreased in NSCLC tissues compared with adjacent normal tissues ($P < 0.0001$).

### 3.2 Both ASO-HOTAIR and si-CCL22 had interference effects

The RT-qPCR results showed that the mRNA expression of HOTAIR of the ASO-HOTAIR group was significantly lower than that of the ASO-NC group (Fig 2A). The mRNA expression of CCL22 of the si-CCL22 group was significantly reduced than that of the si-NC group (Fig 2B) ($P < 0.001$). Compared with the si-NC group, The protein concentration of CCL22 in supernatant decreased significantly ($P < 0.001$) in the si-CCL22 group (Fig 3C). This indicated that the cell models constructed in this study were achieved.

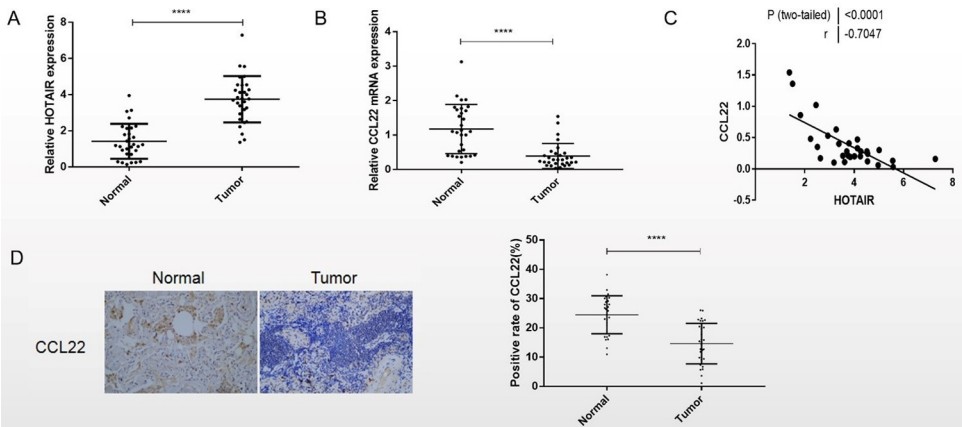

**Fig 1. Clinical correlation between LncRNA HOTAIR and CCL22 in NSCLC examined.** A, B: HOTAIR and CCL22 mRNA expression detected by RT-qPCR. C: The Pearson correlation analysis performed to explore the correlation between HOTAIR and CCL22 in NSCLC. D: The protein expression of CCL22 detected by immunohistochemistry in Cancer tissue and adjacent normal tissue of NSCLC. ****, P<0.001.

## 3.3 The effect of interference with LncRNA HOTAIR and CCL22 on cell function

As shown in Fig 3A, the HOTAIR expression decreased statistically in the ASO-HOTAIR+si-NC group compared with the ASO-NC+si-NC group (P<0.01). There was no statistical difference between the ASO-HOTAIR+si-NC group and the ASO-HOTAIR+si-CCL22 group (P = 0.0621). Compared with the ASO-NC+si-NC, The CCL22 mRNA expression. Increased significantly in the ASO-HOTAIR+si-NC group (P<0.001). The expression of CCL22 mRNA was reduced significantly in the group of ASO-HOTAIR +si-CCL22 compared to ASO-HOTAIR+si-NC (P<0.001) (Fig 3A).

As shown in Fig 3B, the total apoptosis ratio of the ASO-HOTAIR+si-NC group was higher than that of the ASO-NC+si-NC group (6.80% vs. 2.88%). Compared with the ASO-HOTAIR +si-NC group, the total apoptosis ratio decreased in the ASO-HOTAIR+si-CCL22 group (5.41% vs. 6.80%).

From 24h to 72h, cell proliferation increased in all three groups (Fig 3C). At 24h and 48h, there were no significant differences in cell proliferation among the three groups. At 72h, cell proliferation was significantly reduced in the ASO-HOTAIR+si-NC group compared with the

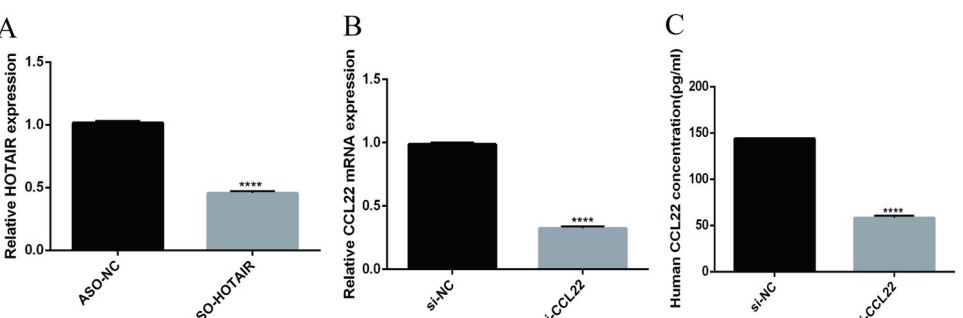

**Fig 2. ASO-HOTAIR suppressing the HOTAIR expression and Si-CCL22 interferring the CCL22 expression.** A, B: the expression of HOTAIR and CCL22 mRNA were detected by RT-qPCR. C: CCL22 protein analyzed by ELISA assay. ****, P<0.001.

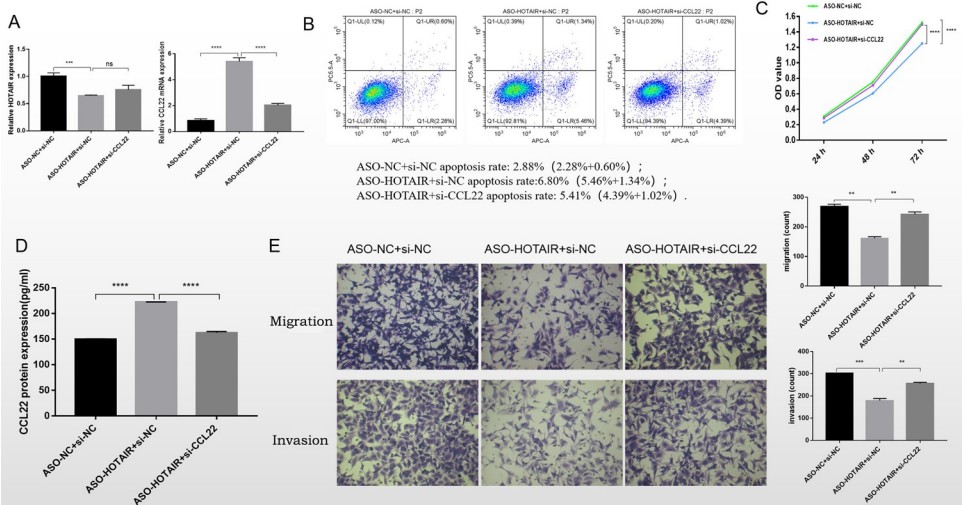

ASO-NC+si-NC apoptosis rate: 2.88% (2.28%+0.60%) ;
ASO-HOTAIR+si-NC apoptosis rate:6.80% (5.46%+1.34%) ;
ASO-HOTAIR+si-CCL22 apoptosis rate: 5.41% (4.39%+1.02%) .

**Fig 3. The inhibition of LncRNA HOTAIR and interference with CCL22 showing the changes in cell function among the ASO-NC+si-NC, ASO-HOTAIR+si-NC and ASO-HOTAIR+si-CCL22 groups.** A: The HOTAIR and CCL22 mRNA expression examined by RT-qPCR B: Cell apoptosis detected by Flow cytometry. C: The cell proliferation tested by MTS assay. D: The CCL22 protein expression tested by ELISA. E: The cell migration and invasion detected by Transwell assay. **, P<0.05; ***, P<0.01; ****, P<0.001;ns, no significance.

group of ASO-NC+si-NC (P<0.01). Then, compared with the group of ASO-HOTAIR+si-NC, cell proliferation increased significantly in the ASO-HOTAIR+si-CCL22 group (P<0.0001).

The CCL22 concentration detected by ELISA is displayed in Fig 3D. Compared with the ASO-NC+si-NC group, the CCL22 protein level increased significantly (P<0.001) in the ASO-HOTAIR+si-NC group. The CCL22 concentration of the ASO-HOTAIR+si-CCL22 group was lower than that of the ASO-HOTAIR+si-NC group (P<0.001).

Pathological images and quantitative analysis results of migration and invasion are shown in Fig 3E. Compared with the ASO-NC+si-NC group, The number of migrating and invading cancer cells was significantly reduced in the ASO-HOTAIR+si-NC group (P<0.05). The number of migrating and invading cancer cells of the ASO-HOTAIR+si-CCL22 group increased significantly than that of the ASO-HOTAIR+si-NC group (P<0.05).

## 4 Discussion

Immune evasion is one of the important reasons for the occurrence, development, and recurrence of NSCLC. Tumor cells evade the recognition and attack of the body's immune system by modifying their surface antigens and changing the tissue microenvironment. Specifically, the balance between immune cells and cytokines in the tumor and peripheral blood is disrupted. Some researchers believe that several marker levels of circulatory inflammation, including chemokines and pro-inflammatory factors, are related to the prospective risk of NSCLC [14]. Chemokines, the ligands of chemokine receptors, are a family of interactive chemotactic cytokines, used to coordinate cell migration and home in the body CCL22 is the ligand of CCR4 transmembrane protein, which is mostly produced by tumor cells and tumor-infiltrating macrophages [15]. However, some researchers have found that in pancreatic and liver cancer, CCL22 is produced by dendritic cells in the tumor, while the cancer cells themselves do not secrete CCL22 in vitro or in vivo [16]. In addition, CCL22 can also be induced in tumor infiltrating immune cells by interleukin-1 (IL-1α) derived from cancer cells [17]. The

regulatory T cells (Tregs) can be attracted by CCL22 into the tumor microenvironment, to reduce anti-cancer immunity [18], which are widely expressed in many cancers, including glioma, gastric cancer, and breast cancer [19–21].

In this study, the expression of CCL22 and LncRNA HOTAIR mRNA in cancer tissues and adjacent normal tissues of NSCLC patients showed significant differences. Compared with adjacent normal tissues, LncRNA HOTAIR was upregulated in NSCLC tissues, while CCL22 mRNA was down-regulated. The immunohistochemical results of CCL22 were consistent with those of RT-QPCR. LncRNA HOTAIR was negatively correlated with CCL22 mRNA expression. CCL22 attracts Treg cells, playing an essential role in the pathogenesis of NSCLC. It was reported that the deregulation of chemokines was associated with the development and progression of many human cancers, including lung cancer. However, the relationships between the survival checkpoint of NSCLC cells and genetic polymorphisms of chemical factors are unclear [22].

In cell experiments, after only interfering with LncRNA HOTAIR, the expression of HOTAIR decreased compared with the control group, while the expression of CCL22 increased in both cells and supernatant, resulting in increased apoptosis, decreased proliferation, and weakened invasion and migration ability. After simultaneous interference with LncRNA HOTAIR and CCL22, the expression of HOTAIR in cells increased compared with that in the LncRNA HOTAIR group, while the expression of CCL22 in cells and supernatant decreased, resulting in decreased apoptosis, increased proliferation, and enhanced invasion and migration. LncRNA HOTAIR, as one kind of LncRNAs, is an oncogene in non-small cell lung cancer. Its possible molecular mechanism and immune regulation pathways involved in NSCLC are still unknown. The tumor microenvironment is a key determinant of tumor progression, which regulates basic cellular processes through different mechanisms [23].

In conclusion, we verified the differential expression of HOTAIR and CCL22 in cancer tissues and adjacent normal tissues in NSCLC patients, and there was a strong negative correlation between HOTAIR and CCL22 mRNA expression. Moreover, the cell experiments demonstrated that LncRNA HOTAIR might promote the proliferation, migration and invasion of the NSCLC cells by inhibiting CCL22 expression. This study is mainly to verify the action mechanism of HOTAIR on NSCLC progression by inhibiting CCL22 expression at the cellular level. Moreover, it needs to be confirmed in the overall models in the future. This is also the direction of future research.

## Supporting information

**S1 Data.**
(ZIP)

## Author Contributions

**Conceptualization:** Hanlin Liang, Jiewen Peng.

**Data curation:** Hanlin Liang, Jiewen Peng.

**Formal analysis:** Hanlin Liang, Jiewen Peng.

**Funding acquisition:** Hanlin Liang, Jiewen Peng.

**Investigation:** Hanlin Liang, Jiewen Peng.

**Methodology:** Hanlin Liang, Jiewen Peng.

**Project administration:** Hanlin Liang, Jiewen Peng.

**Resources:** Hanlin Liang, Jiewen Peng.

**Software:** Hanlin Liang, Jiewen Peng.

**Supervision:** Hanlin Liang, Jiewen Peng.

**Validation:** Hanlin Liang, Jiewen Peng.

**Visualization:** Hanlin Liang, Jiewen Peng.

**Writing – original draft:** Hanlin Liang, Jiewen Peng.

**Writing – review & editing:** Hanlin Liang, Jiewen Peng.

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
