## [Decision Letter · Decision Letter 0]

3 Jan 2022

PONE-D-21-21513

Exploring the mechanisms of the immune regulation by LncRNA HOTAIR and CCL22 signaling pathways about non-small cell lung cancer

PLOS ONE

Dear Dr. Liang,

Thank you for submitting your manuscript to PLOS ONE. After careful consideration, we feel that it has merit but does not fully meet PLOS ONE’s publication criteria as it currently stands. Therefore, we invite you to submit a revised version of the manuscript that addresses the points raised during the review process.

We look forward to receiving your revised manuscript.

Kind regards,

Suhwan Chang

Academic Editor

PLOS ONE

Journal Requirements:

2. In your Methods section, please provide additional details regarding the cell lines used in your study and ensure you have described the source. For more information regarding PLOS' policy on materials sharing and reporting, see https://journals.plos.org/plosone/s/materials-and-software-sharing#loc-sharing-materials, and for more information on PLOS ONE's guidelines for research using cell lines, see https://journals.plos.org/plosone/s/submission-guidelines#loc-cell-lines.

4. Please remove all personal information, ensure that the data shared are in accordance with participant consent, and re-upload a fully anonymized data set. 

Reviewers' comments:

Reviewer's Responses to Questions

**Comments to the Author**

1. Is the manuscript technically sound, and do the data support the conclusions?

Reviewer #1: Partly

Reviewer #2: Yes

2. Has the statistical analysis been performed appropriately and rigorously? 

Reviewer #1: Yes

Reviewer #2: Yes

3. Have the authors made all data underlying the findings in their manuscript fully available?

Reviewer #1: Yes

Reviewer #2: Yes

4. Is the manuscript presented in an intelligible fashion and written in standard English?

Reviewer #1: No

Reviewer #2: Yes

5. Review Comments to the Author

Reviewer #1: Reviewer comments

The authors aimed to explore the mechanism of immune regulation of the long noncoding RNA and CCL22 and their role in the proliferation, migration, and invasion of non-small cell lung cancer. They build upon the existing knowledge and roles of these LncRNA and CCL22 by investigating their expression patterns as well as their effect on apoptosis and migration of NSCLC. Although their experimental procedures used for the various assays are effective, there are a few concerns about the organization of the manuscript

Concerns:

Materials and methods;

The oligonucleotides used in the study and their descriptions should be presented in a tabular form.

The procedure for transfection and maintenance of the cell cultures should be provided in this section.

The experimental procedures for immunohistochemistry, qPCR, and flow cytometry must be detailed in different paragraphs or sub-sections.

The section needs to be revised grammatically and formatted according to the journal's submission requirements to make it an easier read.

Results:

Although the figures and statical analysis are provided in the supporting documents, the authors can state the P-values with the statistical significance on the figures.

Figure 3A can be labeled as 3Ai and 3Aii instead of left and right in the results section.

The font theme and font sizes are not uniform in the manuscript.

The results section is poorly written and undermines the quality of the figures provided in the manuscripts. The results section should be revised.

Minor concerns:

There are numerous grammatical errors in the manuscript that require revisions to make the manuscript easier to read.

Reviewer #2: The manuscript entiltled "Exploring the mechanisms of the immune regulation by LncRNA HOTAIR and CCL22

signaling pathways about non-small cell lung cancer" are well documented. The results are clear abnd well presented.

materials and method are well elaborated.

The following issues need to be addressed.

1. The English need to be improved throughout the manuscript.

2. There are several typing erros. please revise the whole manuscript.

3. The tiltle of manuscript need to improve

4. The existing keywords should be replaced with suitable words, which are not repeated in the title of manuscript

5. In introduction section "Especially, non small cell lung cancer (NSCLC) is the most common histology in lung

cancer, accounting for 80%-85% of the total number of lung cancer". Please put the reference if possible

6. In discussion section, please improve the discussion section with logics.

7. in figures 2 B. please recheck about standared errors

8. please ensure the style of manuscript according to journal standared.

6. PLOS authors have the option to publish the peer review history of their article (what does this mean?). If published, this will include your full peer review and any attached files.

Reviewer #1: No

Reviewer #2: **Yes: **Muhammad Anwar

---

## [Author Response · Author response to Decision Letter 0]

29 Jan 2022

Reply: We've tabulated primer sequences (Table 1).

The procedure for transfection and maintenance of the cell cultures should be provided in this section.

Reply: We have added this sections (Page 5, line 100-113).

The experimental procedures for immunohistochemistry, qPCR, and flow cytometry must be detailed in different paragraphs or sub-sections.

Reply: We have refined the method again (Page 6-8, line 115-163).

The section needs to be revised grammatically and formatted according to the journal's submission requirements to make it an easier read.

Reply: We have asked a professional English teacher to proofread the manuscript in the language. The format of the manuscript was also modified according to the journal.

Results:

Although the figures and statical analysis are provided in the supporting documents, the authors can state the P-values with the statistical significance on the figures.

Reply: We have re-marked the P value in the manuscript.

Figure 3A can be labeled as 3Ai and 3Aii instead of left and right in the results section.

Reply: We have restated the results (Page 9-10, line 186-192). Thank you for your comments!

The font theme and font sizes are not uniform in the manuscript.

Reply: They were done.

The results section is poorly written and undermines the quality of the figures provided in the manuscripts. The results section should be revised.

Reply: We have restated the result sections. Thank you for your comments.

Minor concerns:

There are numerous grammatical errors in the manuscript that require revisions to make the manuscript easier to read.

Reply: We have invited professional teachers to polish the manuscript.

Reviewer #2: The manuscript entiltled "Exploring the mechanisms of the immune regulation by LncRNA HOTAIR and CCL22

signaling pathways about non-small cell lung cancer" are well documented. The results are clear abnd well presented.

materials and method are well elaborated.

The following issues need to be addressed.

1. The English need to be improved throughout the manuscript.

Reply: We have invited professional teachers to polish the manuscript.

2. There are several typing erros. please revise the whole manuscript.

Reply: We have asked the full manuscript to be proofread in the language.

3. The tiltle of manuscript need to improve

Reply: We have rewritten the title (Page 1, line 1-2).

4. The existing keywords should be replaced with suitable words, which are not repeated in the title of manuscript

Reply: The key words have been partially replaced (Page 3, line 47-48).

5. In introduction section "Especially, non small cell lung cancer (NSCLC) is the most common histology in lung

cancer, accounting for 80%-85% of the total number of lung cancer". Please put the reference if possible

Reply: We have added (Page 3, line 54).

6. In discussion section, please improve the discussion section with logics.

Reply: We have added more detail to the discussion section (Page 12, line 250-255; Page 13, line 261-268; Page 13-14, line 273-281).

7. in figures 2 B. please recheck about standared errors

Reply: we have rechecked and revised. Thank you for your comments!

8. please ensure the style of manuscript according to journal standared.

Reply: We have revised the manuscript according to the journal's requirements.

---

## [Editor Report · Decision Letter 1]

2 Feb 2022

LncRNA HOTAIR promotes proliferation, invasion and migration in NSCLC cells via the CCL22 signaling pathway

PONE-D-21-21513R1

Dear Dr. Hanlin Liang,

We’re pleased to inform you that your manuscript has been judged scientifically suitable for publication and will be formally accepted for publication once it meets all outstanding technical requirements.

Kind regards,

Suhwan Chang

Academic Editor

PLOS ONE

Additional Editor Comments (optional):

No further comments
---

## [Editor Report · Acceptance letter]

7 Feb 2022

PONE-D-21-21513R1 

LncRNA HOTAIR promotes proliferation, invasion and migration in NSCLC cells via the CCL22 signaling pathway 

Dear Dr. Liang:

I'm pleased to inform you that your manuscript has been deemed suitable for publication in PLOS ONE. Congratulations! Your manuscript is now with our production department. 

Kind regards, 

on behalf of

Dr. Suhwan Chang 

Academic Editor

PLOS ONE